# OpenReview forum: "Goodtriever: Adaptive Toxicity Mitigation with Retrieval-augmented Models"
_EMNLP/2023/Conference — EMNLP 2023 Findings_

### Official Review · Reviewer_47Kk · 2023-07-24

**Typos Grammar Style And Presentation Improvements:** Ln 172
**Soundness:** 4

**Excitement:**

3: Ambivalent: It has merits (e.g., it reports state-of-the-art results, the idea is nice), but there are key weaknesses (e.g., it describes incremental work), and it can significantly benefit from another round of revision. However, I won't object to accepting it if my co-reviewers champion it.

**Paper Topic And Main Contributions:**

This paper proposes a retrieval-based approach named GOODTRIEVER to mitigate the toxicity problem in text generation.

The key idea is to utilize two external datastores, each containing several key-value pairs annotated as toxic or non-toxic, to aid in controlled generation. Each key-value pair in the datastore represents a pre-computed representation for a context c and a word.

During inference, the model first generates the context representation of the current context, and then compares it with the samples in the datastores.

The model retrieves k-nearest neighbors from the datastore and then calculates a new distribution for the next token via Eq. 4, which is then fused with the normal distribution generated by the LM. The word probabilities are rewarded if they exist in the non-toxic datastore and penalized otherwise.

**Reasons To Accept:**

* This paper offers a new method to mitigate toxicity problem in text generation. The biggest advantage, as claimed by the authors, is that the proposed method is training-free, which can be well generalized to other domains and alliviate the so-called continual toxicity problem to some extent.

**Reasons To Reject:**

* The novelties are limited. As discussed in related works, the idea is closely related to the approaches in retrieval-augmented LMs, and also reminiscent of traditional instance-based methods, e.g., [1].
* I take a cautious stance on the advantages of latency reduction claimed by the author. I can see in B.3 that the authors use quite large datastores for retrieval-augmentation. Does this incur a severe repr storage burden and computational burden of the kNN process, especially considering that the generation works in an auto-regressive fashion, i.e., retrieving needed samples one by one? Maybe I misunderstand sth. I would be glad to change my score if the authors make a clarification.

[1] Instance-Based Neural Dependency Parsing

**Reproducibility:**

4: Could mostly reproduce the results, but there may be some variation because of sample variance or minor variations in their interpretation of the protocol or method.

**Reviewer Confidence:**

3: Pretty sure, but there's a chance I missed something. Although I have a good feel for this area in general, I did not carefully check the paper's details, e.g., the math, experimental design, or novelty.

---

> ### Author Rebuttal · Authors · 2023-08-28
>
> We are grateful for the reviewer’s **47Kk** constructive feedback, and for acknowledging that our “training-free” method “can be well generalized to other domains” and “alleviate the so-called continual toxicity problem to some extent”.
>
> > **The novelties are limited. As discussed in related works, the idea is closely related to the approaches in retrieval-augmented LMs, and also reminiscent of traditional instance-based methods, e.g., [1].**
>
> There are two main novelties of our work: **(1) the usage of multiple datastores as a product of experts (PoE) and (2) the possibility of dynamic toxicity mitigation through datastore updates and no model training.** We understand these are considerable contributions to both the retrieval-augmented and toxicity-mitigation fields of study, respectively. (2) Is particularly critical given the ever-evolving nature of both language and toxic text. We are not aware of any prior work which has made these contributions.
>
> Previous works have relied on a single datastore to guide generations toward an aimed distribution. We, on the other hand, guide generations both toward what we want (non-toxic) and what we don’t want (toxic). _To the best of our knowledge, we are the first to steer the model away from a datastore distribution, as well as the first to apply PoE with datastore-generated probabilities._
>
> Similarly, _we are the first to use retrieval-augmentation to control for the toxicity attribute_. The usage of datastores provides intrinsic explainability capabilities, a critical feature for the user. In our experiments from Section 4, we showcase the adaptiveness of our technique by continually adding domains and reducing the model’s toxicity.
>
>
> > **I take a cautious stance on the advantages of latency reduction claimed by the author. I can see in B.3 that the authors use quite large datastores for retrieval-augmentation. Does this incur a severe repr storage burden and computational burden of the kNN process, especially considering that the generation works in an auto-regressive fashion, i.e., retrieving needed samples one by one? Maybe I misunderstand sth. I would be glad to change my score if the authors make a clarification.**
>
> First of all, thank you for providing us with the opportunity to clarify this topic. We hope the following explanation is sufficient to establish our kNN retrieval method as computationally efficient, and that it could be further improved through dedicated effort.
>
> **Datastore size.** The datastore size from our work is actually quite small when compared to other retrieval-augmentation works. In kNN-LM [1], for example, they use a datastore of 3 billion tokens, while other works [2] have up to a trillion tokens in their datastore. _**Our bigger datastore (non-toxic), on the contrary, contains over 40 million tokens, while the toxic contains about 9 million tokens. That corresponds to, respectively, about 1.33% and 0.3% of the tokens of kNN-LM’s datastore [1]**_. We also show in Figure 2 how we can make the datastores significantly smaller and achieve similar results.
>
> **The computational burden of the kNN-search process.** In our implementation, we make use of the FAISS library for GPU-backed similarity search operations. In one of their papers [3], the group mentions that their algorithm “operates at up to 55% of theoretical peak performance, enabling a nearest neighbor implementation that is 8.5× faster than prior GPU state of the art”.
>
> Following previous work [4], we use half-precision (fp16) to store datastore keys and also use a quantized index (IndexIVFPQ class), which means we actually perform an **approximate similarity search**. In this type of index, index vectors are clustered with k-means, and similarity operations are performed primarily according to cluster centroids. **Quantized indexes are significantly faster than performing one-by-one similarity operations**, which would be the case for the so-called “flat index”, at a cost of approximate results.
>
> In Table 2 we observe how Goodtriever adds 77% of inference time cost on top of GPT2 large base model while performing the search operation twice (for toxic and non-toxic datastores). That is a significant reduction over the 212% added by DExperts expert models.
>
> We believe that with dedicated effort these computation metrics can be improved, as shown in previous work [5, 6]. We leave further optimizations for future work.
>
> We hope this answer quells your concerns about the computation efficiency of retrieval methods.
>
>
> [1] Khandelwal, Urvashi, et al. "Generalization through memorization: Nearest neighbor language models." arXiv preprint arXiv:1911.00172 (2019).
>
> [2] Borgeaud, Sebastian, et al. "Improving language models by retrieving from trillions of tokens." International conference on machine learning. PMLR, 2022.
>
> [3] Johnson, Jeff, Matthijs Douze, and Hervé Jégou. "Billion-scale similarity search with gpus." IEEE Transactions on Big Data 7.3 (2019): 535-547.
>
> [4] Alon, Uri, et al. "Neuro-symbolic language modeling with automaton-augmented retrieval." International Conference on Machine Learning. PMLR, 2022.
>
> [5] Izacard, Gautier, et al. "A memory efficient baseline for open domain question answering." arXiv preprint arXiv:2012.15156 (2020).
>
> [6] Min, Sewon, et al. "Neurips 2020 efficientqa competition: Systems, analyses and lessons learned." NeurIPS 2020 Competition and Demonstration Track. PMLR, 2021.

---

### Official Review · Reviewer_9NcL · 2023-08-03

**Soundness:** 4

**Excitement:**

4: Strong: This paper deepens the understanding of some phenomenon or lowers the barriers to an existing research direction.

**Justification For Ethical Concerns:**

The authors already state possible limitations or risks of their work. Such risks are the same as previous methods proposed for this task.

**Paper Topic And Main Contributions:**

The paper proposes a new toxicity mitigation method that allows the use of external databases to rescore LM's next token probabilities. A particularity of this work is being able to combine several datastores during the knn-decoding phase. Several experiments are proposed, including different datastore sizes, showing that the proposed method is able to perform toxicity mitigation similarly to previous state-of-the-art methods while being more efficient in terms of computation and number of parameters.

**Questions For The Authors:**

- While perplexity is not significantly changed, modifying the tokens during inference can affect the context information that the model can use. Have you performed experiments on probing tasks to measure the model's performance?
- Your datastores come from human-annotated datasets. Could this method be applied to automatically annotated data?


**Reasons To Accept:**

- The paper is well written, and the method applied is easy to follow.
- Experimental results are convincing and clearly showcase the effectiveness of the method.
- Toxicity mitigation methods are known for slowing inference significantly. Efficient methods are a  research area worth exploring.

**Reasons To Reject:**

- The paper lacks some results on the performance of the models on other tasks. While perplexity does not increase significantly, it would be interesting to know the impact of the method on other probing tasks.
- A limitation of the method is its dependency on human-annotated datastores. The paper lacks some results on the quality of the datastores data on the final performance of the model.

**Reproducibility:**

4: Could mostly reproduce the results, but there may be some variation because of sample variance or minor variations in their interpretation of the protocol or method.

**Reviewer Confidence:**

4: Quite sure. I tried to check the important points carefully. It's unlikely, though conceivable, that I missed something that should affect my ratings.

---

> ### Author Rebuttal · Authors · 2023-08-28
>
> We thank the reviewer **9NcL** for the meaningful comments and for saying that “the paper is well written, and the method applied is easy to follow.” We are thrilled that you find our “experimental results are convincing and clearly showcase the effectiveness of the method”.
>
> > **A limitation of the method is its dependency on human-annotated datastores. The paper lacks some results on the quality of the datastores data on the final performance of the model.**
>
> > **Your datastores come from human-annotated datasets. Could this method be applied to automatically annotated data?**
>
> Yes, it can be applied to automatically annotated data. We chose to experiment mainly with the Jigsaw datastore, since it is a large human-annotated dataset commonly used in toxicity mitigation and classification studies.
>
> During the rebuttal period, we performed additional experiments to demonstrate the robustness of Goodtriever by substantially reducing the size of the datastores and automatically annotating them. We perform such experiments with 2 datasets as datastores: Jigsaw, our main dataset, and RealToxicityPrompts (RTP). Base models are kept the same, and so are generation parameters described in Appendix B.4.
>
> Respectively for toxic and non-toxic datastores, **reported experiments have about 16x and 40x smaller datastores than the results shown in Table 1**.
>
> **In the table below** we show results of Goodtriever with _**substantially smaller automatically annotated datastores by Perspective API**_.  Surprisingly, at this data-constraint regime, **both variants of automatically-labeled Goodtriever datastores (Jigsaw and RTP) achieve lower toxicity metrics** than the variant with a full-sized human-annotated Jigsaw from the paper’s Table 1. Due to the smaller toxic datastore (i.e. Figure 2 from the paper), diversity is slightly lower for both variants. It is also remarkable how Goodtriever with the randomly subsampled human-annotated Jigsaw performs on par with its much larger version from Table 1.
>
> We thank the reviewer for their feedback and the opportunity to run additional experiments which we believe has made our work stronger, _we are committed to adding these details and experiments to the final manuscript_. We hope such results have eased your concern regarding data dependency and showed that Goodtriever effectively reduces toxicity in generations even with automatically labeled data and significantly smaller datastores.
>
> | Datastore | Human Annotated | EMT | TP | Perplexity | Diversity (dist-1) | Toxic Tokens | Non-Toxic Tokens |
> |---|---|---|---|---|---|---|---|
> | RTP (new experiment) | No | 0.19 | 0.02 | 23.31 | 0.52 | 645k | 808k |
> | Jigsaw (new experiment) | No | 0.18 | 0.03 | 29.47 | 0.55 | 600k | 900k |
> | Jigsaw (new experiment) | Yes | 0.22 | 0.04 | 29.92 | 0.57 | 640k | 857k |
> | Jigsaw (Table 1) | Yes | 0.22 | 0.04 | 27.11 | 0.58 | 9.4M | 41.7M |

---

### Official Review · Reviewer_NexP · 2023-08-05

**Soundness:** 3

**Excitement:**

3: Ambivalent: It has merits (e.g., it reports state-of-the-art results, the idea is nice), but there are key weaknesses (e.g., it describes incremental work), and it can significantly benefit from another round of revision. However, I won't object to accepting it if my co-reviewers champion it.

**Paper Topic And Main Contributions:**

This paper propose a retrieval-based toxic generation mitigation method, which takes into account the toxicity's change nature and works in decoding time. This method obtains a comparable performance with the current SOTA toxicity mitigation while achieving 43% relative latency reduction during inference and being more computationally efficient. Besides, this method enables controlled text generation.

**Questions For The Authors:**

1. Could you compare your methods with other baseline models in terms of inference time and parameters?
2. How the datastores are constructed? Could you give me a detailed description?

**Reasons To Accept:**

1. This method achieves a comparable performance with SOTA methods but reducing the inference time by 43%.
2. This method is evaluated at different model sizes and families, and results show that this method remains efficient in mitigating toxicity even as the model size increases.
3. This method achieves competitive performance compared to the baseline models when it is utilized to mitigating toxicity  newly added domain.

**Reasons To Reject:**

1. There is still a lot of room for improvement in the task performance of the model.
2. The evaluation metric for toxicity may not be so reliable. The Pespective API ofen makes mistakes.
3. This method's performance greatly relies on the datastores constructed from the datasets.

**Reproducibility:**

4: Could mostly reproduce the results, but there may be some variation because of sample variance or minor variations in their interpretation of the protocol or method.

**Reviewer Confidence:**

3: Pretty sure, but there's a chance I missed something. Although I have a good feel for this area in general, I did not carefully check the paper's details, e.g., the math, experimental design, or novelty.

---

> ### Author Rebuttal · Authors · 2023-08-28
>
> We thank reviewer **NexP** for their feedback and for acknowledging our method “achieves a comparable performance with SOTA methods but reducing the inference time by 43%” and that our “method remains efficient in mitigating toxicity even as the model size increases”.
>
>
> > **There is still a lot of room for improvement in the task performance of the model.**
>
> We understand [**NexP**]’s comment as they would like to completely mitigate toxicity in generations. We also strive for that, but as they acknowledged, _**our method “achieves a comparable performance with SOTA methods”**_.
>
> We also emphasize that _Goodtriever is able to **reduce the absolute number of toxic continuations more than 10 times**_. If we look at the Toxic Fraction [1] (average of the number of toxic continuations per prompt divided by the number of total continuations per prompt), which corresponds to the fraction of instances that are classified as toxic. The Toxic Fraction for Goodtriever is 0.2%. In contrast, in our base model GPT2-Large without Goodtriever, the toxic fraction is 2.33%. _Goodtriever **reduces the Toxic Fraction from 2.33% to 0.2%**_.
>
> [1]  Liang, Percy, et al. "Holistic evaluation of language models." arXiv preprint arXiv:2211.09110 (2022).
>
>
> > **The evaluation metric for toxicity may not be so reliable. The Perspective API often makes mistakes.**
>
> We agree with the reviewer – as we point out in our Limitations section, we understand there is not an exclusive right answer for what is toxic or not, as the perception of toxicity largely depends on cultural and societal aspects of the annotator. Similarly, as no evaluation metric is perfect, Perspective API also has its strengths and weaknesses. **This API is the most widely used in the field, with extensive research on its limitations [2, 3]. We chose to adopt not only for coherence with the best automatic evaluation standards and ease of comparability to other work, but mainly because we know of its limitations, which is not true for other toxicity detection tools.** All our baselines have used this tool to measure toxicity, and it is also leveraged to assess model risk by prestigious models and benchmarks such as PaLM2 [4] and HELM [1], respectively.
>
> [2] Pozzobon, Luiza, et al. "On the Challenges of Using Black-Box APIs for Toxicity Evaluation in Research." arXiv preprint arXiv:2304.12397 (2023).
>
> [3] Rauh, Maribeth, et al. "Characteristics of harmful text: Towards rigorous benchmarking of language models." Advances in Neural Information Processing Systems 35 (2022): 24720-24739.
>
> [4] Anil, Rohan, et al. "Palm 2 technical report." arXiv preprint arXiv:2305.10403 (2023).
>
>
>
> > **This method's performance greatly relies on the datastores constructed from the datasets.**
>
> We believe our method is robust to varying the quality and quantity of data in the datastore.
> During the rebuttal period, we performed additional experiments to demonstrate this robustness by substantially reducing the size of the datastores and automatically annotating them. We perform such experiments with 2 datasets as datastores: Jigsaw, our main dataset, and RealToxicityPrompts (RTP). Base models are kept the same, and so are generation parameters described in Appendix B.4.
>
> Respectively for toxic and non-toxic datastores, **reported experiments have about 16x and 40x smaller datastores than the results shown in Table 1**.
>
> **In the table below** we show results of Goodtriever with _**substantially smaller automatically annotated datastores by Perspective API**_.  Surprisingly, at this data-constraint regime, **both variants of automatically-labeled Goodtriever datastores (Jigsaw and RTP) achieve lower toxicity metrics** than the variant with a full-sized human-annotated Jigsaw from the paper’s Table 1. Due to the smaller toxic datastore (i.e. Figure 2 from the paper), diversity is slightly lower for both variants. It is also remarkable how Goodtriever with the randomly subsampled human-annotated Jigsaw performs on par with its much larger version from Table 1.
>
> We thank the reviewer for their feedback and the opportunity to run additional experiments which we believe has made our work stronger, _we are committed to adding these details and experiments to the final manuscript_. We hope such results have eased your concern regarding data dependency and showed that Goodtriever effectively reduces toxicity in generations even with automatically labeled data and significantly smaller datastores.
>
>
> | Datastore | Human Annotated | EMT | TP | Perplexity | Diversity (dist-1) | Toxic Tokens | Non-Toxic Tokens |
> |---|---|---|---|---|---|---|---|
> | RTP (new experiment) | No | 0.19 | 0.02 | 23.31 | 0.52 | 645k | 808k |
> | Jigsaw (new experiment) | No | 0.18 | 0.03 | 29.47 | 0.55 | 600k | 900k |
> | Jigsaw (new experiment) | Yes | 0.22 | 0.04 | 29.92 | 0.57 | 640k | 857k |
> | Jigsaw (Table 1) | Yes | 0.22 | 0.04 | 27.11 | 0.58 | 9.4M | 41.7M |
>
>
>
> > **​​Could you compare your methods with other baseline models in terms of inference time and parameters?**
>
> In our original manuscript, we benchmark the latency of Goodtriever against DExperts using the same hardware, and codebase to ensure a fair comparison. **We find that Goodtriever is 47% faster than DExperts in our evaluations**. In addition, we append Table 14 from DExperts [5] below, which could be used as a proxy comparison with Goodtriever, as they report the same metric as us (the number of seconds to generate 20 tokens). _Given the margin of 47% by which we outperform DExperts, we have reason to believe that our method is also significantly faster than those other models as well._
>
> Regrettably, to ensure a fair comparison of inference times, we would need to completely replicate the techniques we are benchmarking against within our codebase. This replication would allow us to perform benchmarks on identical hardware. Due to the constraints of the limited time frame of the rebuttal process, achieving this level of replication is not feasible. We hope the extra information provided by the appended table is sufficient to ease your concerns.
>
>
> | Model | Generation time (sec) |
> |---|---|
> | GPT-2 / DAPT | 0.094 |
> | DExperts (small) | 0.186 |
> | DExperts (medium) | 0.240 |
> | DExperts (anti-only) | 0.248 |
> | GeDi | 0.276 |
> | DExperts (large) | 0.334 |
> | PPLM | 25.39 |
>
> [5] Liu, Alisa, et al. "DExperts: Decoding-time controlled text generation with experts and anti-experts." arXiv preprint arXiv:2105.03023 (2021).
>
>
> > **How the datastores are constructed? Could you give me a detailed description?**
>
> Sure! You can also find a more rigorous description in Section 2.1 of the paper.
>
> Datastores are constructed by forward-passing every sample from dataset _D_ into a function _f_. In our case, that function _f_ is a model _M_. Simplifying to a case where each word corresponds to a token, a sentence such as “She is a great person” yields 5 samples. The first sample is composed of the _(context, next-token)_ pair _(“She”, “is”)_. In our notation, that is $(c_i, w_i)$. The context $c_i$ “She” is passed through _M_ and we get the context’s embedding from the last linear layer of _M_. The datastore is composed of _(key, value)_ pairs of _(context embedding, next-token id)_, referenced as $(k_i, v_i)$ in the paper. In the sequence we’re analyzing, the 5th sample would be obtained from (“She is a great”, “person”). The datastore size is computed by the number of total samples obtained in the described manner.

---

### Meta-Review · Area_Chair_eEb7 · 2023-09-19

**Recommendation:** 3

**Metareview:**

The paper proposes a toxicity mitigation method that takes into account the changing nature of the text. The premise of the adaptable mitigation technique is commendable. The main contribution of the work is a reduction in inference time by 43% and minimizing computational requirements. The contribution is rather incremental but there are no critical issues that are unresolved after the rebuttal. I would suggest the authors to add the additional experiments reported in the final version of the paper.

---

### Decision · Program_Chairs · 2023-10-07

**Decision:**

Accept-Findings

**Comment:**

The paper proposes a toxicity mitigation method that takes into account the changing nature of the text. The premise of the adaptable mitigation technique is commendable. The main contribution of the work is a reduction in inference time by 43% and minimizing computational requirements. The contribution is rather incremental but there are no critical issues that are unresolved after the rebuttal. I would suggest the authors to add the additional experiments reported in the final version of the paper.